# Theoretical Prediction of Structures and Properties of 2,4,6-Trinitro-1,3,5-Triazine (TNTA) Green Energetic Materials from DFT and ReaxFF Molecular Modeling

**DOI:** 10.3390/ma15113873

**Published:** 2022-05-29

**Authors:** Ming-Ming Zhou, Dong Xiang

**Affiliations:** College of Chemistry and Environmental Engineering, Yangtze University, Jingzhou 434023, China; 2021720391@yangtzeu.edu.cn

**Keywords:** crystal morphology, cell parameters, intermolecular interaction, Hirshfeld surface, pyrolysis mechanism

## Abstract

Nitryl cyanide, O_2_NCN, as a new high-energy molecule, has not yet been successfully synthesized. It has prompted us to conduct a theoretical study of its possible space structures and properties. The RESP charges and the most stable spatial structures demonstrate that crystal morphology is affected by both the main nonbonded interactions and the molecular arrangement. The crystal structure prediction indicated that there are seven structures, namely P1, P2_1_, P2_1_2_1_2_1_, P2_1_/c, Pna2_1_, Pbca, and C2/c. The most stable space structure is likely to be Pna2_1_ and the corresponding cell parameters are Z = 4, a = 8.69 Å, b = 9.07 Å, c = 9.65 Å, and α = β = γ = 90.0°. To further study the intermolecular interactions of TNTA, a series of theoretical analyses were employed, including Hirshfeld surface analysis and fingerprint plots. The pyrolysis mechanism and properties show that high temperatures can promote decomposition. The systematic search approach can be a new strategy to identify structures effectively and has the potential to provide systematic theoretical guidance for the synthesis of TNTA.

## 1. Introduction

The ab initio theory was used to assess the energy performance and other important properties of 2,4,6-trinitro-1,3,5-triazine (TNTA) in 1907 [1]. The explosive performance of TNTA is superior to that of cyclotetramethylene tetranitramine (HMX), and it has a higher density of 2.1 g·cm^−3^ [2]. Korkin, A. A. et al. [3] conducted a theoretical study of its possible derivatives with similar performance (exothermic decomposition) but with possibly increased stability and a higher density in its condensed state. The research found that the higher the stability of the six-membered ring structure, the higher the density and energy released when it decomposes into a stable species, determining that TNTA is a potential high-energy compound. Yang K. [4] studied potential synthetic routes using the MP2/6-311G(d,p)//B3LYP/6-31G(d) levels of theory. Values of the heat of formation in the solid phase were predicted from density functional theory calculations. Densities were estimated from a regression equation obtained by molecular surface electrostatic potentials for TNTA. This work also suggested that TNTA might be formed in a solution if the trimerization reaction is carried out in a concentrated solution of nitryl cyanide. However, TNTA was not successfully synthesized. This current study rapidly predicted the structure and pyrolysis mechanism of TNTA, which can provide theoretical guidance for the experimental preparation.

The strength of intermolecular and intramolecular nonbonded interactions can be measured by restrained electrostatic potential (RESP) [5,6,7]. Zhang et al. [8,9,10,11,12] provided a detailed summary of the intermolecular interactions that play a dominant role in the packaging structures. Predicting crystal structures remains a challenging problem [13]. Some studies [14,15] have predicted the structures and properties of high-energy materials. Other studies [16,17,18,19] have studied the thermal decomposition mechanism of high-energy materials by ReaxFF reactive molecular dynamics simulations. In this study, firstly, partial RESP charges and unit cell parameters were used to predict the space group and packing of TNTA. Then, Hirshfeld surface analysis and fingerprint plots were used to study the molecular interactions of TNTA. Finally, the thermal decomposition of TNTA was researched by reactive molecular dynamics using ReaxFF-lg.

## 2. Computational Details

The RESP charges were generated as follows: All molecules were optimized at the B3LYP/6-311g(d,p) density functional theory (DFT) level using Gaussian 16 software [20]. Then, the RESP charges were fitted from the optimized geometry and wave function using Multiwfn software [21]. The RESP charges of the monomers were derived from the respective trimers optimized by DFT, which is similar to the reported treatment method [22].

The Hirshfeld surfaces [23] or isosurfaces of the electron density were mapped by CrystalExplorer [24], which is a freely available software. Two-dimensional mapping [25,26], a simple color plot, was used to analyze the intermolecular interactions quantitatively and qualitatively. The distances to the nearest atoms outside, de, and inside, di [26,27,28], were defined as the points of the intermolecular contacts.

The ReaxFF-lg force field using the LAMMPS program package performed all the molecular simulations. The original cell structure was a CIF file downloaded from CCDC, and the single-crystal cell was expanded into 3 × 4 × 2 supercells as the research substrate for dynamic simulation. The molecular formula and atomic numbering of TNTA are shown in Figure 1. First, the canonical ensemble (NVT) and the Berendsen thermostat were applied to the molecular dynamics (MD) simulation with a total time of 10 ps at 300 K, which further relaxed the TNTA supercell. Then, ReaxFF-lg isothermal–isochoric MD (NVT-MD) simulations were performed for 300 ps at 1800, 2250, 2500, 3000, and 3500 K, respectively, controlled by the Berendsen thermostat based on the relax supercell. An analysis of the fragments was performed with a 0.3 bond order cutoff value for each atom pair to identify the chemical species. The information of the dynamic trajectory was recorded every 50 fs, which was used to analyze the evolution details of TNTA in the pyrolysis process.

Simulated annealing of the global minimum energy, a thermodynamic problem, was performed by Monte Carlo (MC). The crystal was heated quickly to a high temperature and then cooled slowly to obtain the annealing structure. The packing groups were modified by the cell parameters by rotating and translating the rigid molecular units. For each space group, the low potential energy (E) packings were selected.

## 3. Results and Discussions

### 3.1. Crystal Structure Predict

The detailed COMPASS force field parameters and partial RESP charges are given in Table 1. The partial RESP charge of C1–C3 was 0.78. Meanwhile, the partial RESP charge of N1–N3 was −0.73. The greater the difference in the electronegativity between C1–C3 and N1–N3, the stronger the stabilizing bond–antibond interactions [5,6]. This demonstrates that the benzene ring-like structure is much more stable than other bonds. Moreover, the nonbonds of (C1–C3)/(N1–N3) have a significant effect on the intermolecular stacking [7]. Both the partial RESP charges of C1–C3 and N4–N6 were positive. The difference in the partial RESP charges between C1–C3 and N1–N3 was exactly the same as the value of the NO_2_ fragment. This demonstrates that the bonds of C1–C3 and N4–N6 are relatively weak. The bonding of C1–C3 and N4–N6 is mainly due to the interaction between N4–N6 and O1–O6. This nonbonded interaction has an effect not only on the intramolecular but also on the intermolecular stacking. The nonbonded interactions of (C1–C3)/(N1–N3) and (N4–N6)/(O1–O6) determine the stacking of molecules.

The impact sensitivity (IS) could largely be affected by the crystal packing structure [5,6,7]. Therefore, we investigated the variations in the crystal packing structures from a viewpoint of seven space groups, as shown in Figure 2. The space groups P2_1_2_1_2_1_, P2_1_/c, and C2/c are not affected by the intermolecular interactions on molecular stacking. So, they are not the most reasonable spatial arrangement. The P1 and P2_1_ space groups are affected by the nonbonded interaction of (N4–N6)/(O1–O6). However, the strongest nonbonded interaction of (C1–C3)/(N1–N3) is ignored. The packing structure of the Pbca space group is the exact opposite to that of the P1 and P2_1_ space groups. The random molecular arrangement is not conducive to impact sensitivity. The packing structure of Pna2_1_ combines the main nonbonded interaction, as shown in Table 1, and the graphene-like structure of TATB [29]. The Pna2_1_ space group is the most likely spatial arrangement.

We employed the COMPASS force field and the Polymorph module in Materials Studio (MS) 4.4 2008 to obtain the possible molecular packing in the crystal state. Statistical data [30,31,32,33] demonstrate that most crystals belong to seven space groups (P1, P2_1_, P2_1_2_1_2_1_, P2_1_/c, Pna2_1_, Pbca, and C2/c), and the global search was confined to these groups only. The space group with the lowest energy was selected as the most likely molecular packing. The input structure for the polymorph search came from the ground-state geometry calculation at the B3LYP/6-311G(d,p) level. Here, we applied the PBE and PW91 pseudopotentials to the GGA and the CA-PZ pseudopotential to the LDA functional to the input structures. Table 2 shows the packing energy and cell parameters of the seven space groups. It was found that the differences among the packing energies estimated by the GGA-PBE for the seven space groups were much smaller than those estimated by the GGA-PW91 and LDA-CA-PZ. The energies were in the range of −4.24 to −0.87 kJ·mol^−1^ and the Pna2_1_ space group had the lowest energy. The lower the lattice energy, the more stable the crystal structure. Therefore, due to the lowest Gibbs free energy, the Pna2_1_ space group is the most likely crystal structure for TNTA. The corresponding lattice parameters were Z = 4, a = 8.69 Å, b = 9.07 Å, c = 9.65 Å, and α = β = γ = 90.0°.

To obtain a better understanding of the crystal stacking, the intermolecular interactions [11,34] of single crystals were studied through Hirshfeld surface analysis using freeware [26,35]. In the Hirshfeld surface analysis, the red and blue areas represent the high and low probabilities of close contact with external molecules, respectively [36]. Two features can be seen on the surfaces in Figure 3. One is that the interaction takes place through both the external and internal atoms because the red dots are distributed on both the front and side faces. These irregular surfaces show that TNTA molecules are irregular, more uneven, and less planar. The spatial symmetry of the whole surface shape of Pna2_1_ is relatively better than the other space groups. This shows that the crystal stacking of Pna2_1_ is more spatially symmetrical. The other feature is that the red dots are concentrated around the oxygen atoms. O···O contacts become dominant in TNTA molecules. The more oxygen atoms exposed on the exterior of TNTA molecules, the more sensitive the molecules are. Figure 4 shows that the O···O contact percentage of the Pna2_1_ space group is 39%, which is the lowest percentage of the seven space groups. These results demonstrate that the Pna2_1_ space configuration is the most stable crystal stacking for TNTA molecules.

The percentage of N···O contacts is the second highest among the seven space groups, at 43%, as shown in Figure 4. The C···O contacts are the other main intermolecular interactions. Looking at Figure 1 and Figure 3, we can draw the conclusion that N···O and C···O contacts may contribute to planar conjugated molecular structures between the NO_2_ group and the heterocyclic group. This may be because the crystal stacking of Pna2_1_ is much more stable than the others.

Two-dimensional plots of these intermolecular contacts are shown in Figure 5. The narrow orange line denoting O···O contacts is much more obvious in the plot of the P1 space group, which suggests an increase in the O···O contacts. This demonstrates that the crystal stacking is much more sensitive in the P1 space group.

### 3.2. Reactive Molecular Dynamics Using ReaxFF-lg

The RESP charges, space groups, and intermolecular interactions predicted that Pna2_1_ has a much more stable crystal morphology. Therefore, the MD simulations were performed solely for the crystal structure of the Pna21 space group. The evolution of the potential energy (PE) of the system is shown in Figure 6. The curves, except the 1800 K curve, first decrease steadily, then reach an approximately horizontal line, indicating that TNTA breaks down rapidly and instantaneously at these extreme temperatures. The curve of 1800 K decreases steadily, implying that TNTA is not completely decomposed within 300 ps. The asymptotic value of PE increased with the temperature increase. The declining rate of PE demonstrates the accelerating release of heat as the temperature increased.

The population of TNTA molecules reduced rapidly under five extreme conditions as shown in Figure 7. With increases in temperature, the population of the TNTA molecules disappeared faster. This implies that increasing the temperature can accelerate the decomposition rate of TNTA.

Figure 8 demonstrates the time evolution of the main fragments of NO_2_ during the whole decomposition stage under the five extreme conditions. All the curves, except the 1800 K curve, initially rapidly increase and then decrease until they disappear over time, indicating that TNTA first breaks the C-NO_2_ bond to generate NO_2_, and then all of the NO_2_ fragments decompose into more stable products, such as N_2_, over time. As the temperature increased, the maximum value of the NO_2_ and the rates of increase and decrease in the amount of NO_2_ were higher, indicating that increases in temperature can accelerate the decomposition rate of TNTA. At 1800 K, the amount of NO_2_ first increased and then oscillated around the equilibrium value of 30. The results show that TNTA does not decompose completely in 300 ps. This conclusion is consistent with the results in Figure 6 and Figure 7.

The curves of the amount of NO initially rapidly increase and then decrease until reaching an equilibrium value, indicating that TNTA first converts nitro to nitroso, and then a part of the NO fragments decompose into more stable products over time. As the temperature increased, the maximum value of NO and the rate of increase in the amount of NO were higher, indicating that increases in the temperature can accelerate the decomposition rate of TNTA. However, the equilibrium value was lower with the temperature increase, demonstrating that higher temperatures can accelerate NO fragment decomposition into more stable products.

The curves of the amount of NO_3_ initially slightly increase and then decrease with time evolution. As the temperature increased, the maximum value of NO_3_ and the rate of increase in the amount of NO_3_ was lower, indicating that increases in the temperature can inhibit the production of NO_3_. The NO_3_ disappeared, except at 1800 K, it reached an equilibrium value. This demonstrates that higher temperatures can accelerate the complete decomposition of TNTA.

The decay rate of TNTA molecules increased with increasing temperature. The evolution of the amounts of the final products (N_2_, and CO_2_) over time is shown in Figure 9. The final products were constantly produced, and their amounts were nearly stable at the end. The N_2_ and CO_2_ increased sharply, then reached equilibrium, except at 1800 K. However, the value of N_2_ and CO_2_ increased steadily over 300 ps. This demonstrates that TNTA decomposes totally except at 1800 K. The equilibrium values were larger as the temperature increased. It took less time to reach equilibrium as the temperature increased. This implies that higher temperatures can accelerate the decomposition of TNTA. An interesting phenomenon is that the variation trend of N_2_ is consistent with that of CO_2_. At the same time, the amount of N_2_ was slightly higher than the amount of CO_2_ during the whole reaction time at all five high temperatures.

For each temperature, the system was heated to the target temperature within 0.3 ps and the C-NO_2_ bond was broken to release NO_2_ fragments. The life time was shorter as the temperature increased and the maximum number of NO_2_ fragments was larger with the temperature increase, indicating that higher temperatures would promote the production of NO_2_ and then promote its decomposition into more stable products. The appearance time of the fragments NO, NO_3_, CO_2_, and N_2_ became shorter as the temperature increased. This demonstrates that a high temperature will accelerate the decomposition of TNTA. The maximum amounts of NO_2_, NO, CO_2_, and N_2_ increased with the temperature increase. However, for NO_3_, the change rule was just the opposite. These results imply that higher temperatures can better promote the production of stable products.

Table 3 demonstrates the relatively higher NF of the reactions at 2500 K and 3500 K. We selected the two temperatures (3500 K, the higher in the series) to illustrate the main trends because of the higher number of reactions.

One of the major intermediate species of the maximum net flux of TNTA is C_2_O_4_ (2CO_2_ → C_2_O_4_ (R2)), corresponding to reaction 61 after reversible reaction just shown as Table 4. Another major intermediate is C_3_O_5_ and this molecule appeared at CO_2_ + C_2_O_3_ → C_3_O_5_ (R12), corresponding to reaction 15. From the global point of view, we observed two main trends: (a) the important reactions of CO_2_ abstraction (R3, 4, 6, 12, 16, 18, 21, 26, 29, 30) that occurred due to the high concentration of CO_2_; (b) the breakdown of fragments generating CO_2_ (R1, 2, 5, 7, 15, 17, 22, 23, 24, 25, 26, 27) that occurred due to the stability of CO_2_. Conversely, the overall analysis of simulations shows that the other main gas product is N_2_. The chemical effect of N_2_ can be expressed in terms of two types of reactions: R-C-N_2_ → N_2_ + R-C (R1, 9, 10, 20, 28) and R-N-N_2_→ N_2_ + R-N (R8, 13). Thus, in general, increases in CO_2_ and N_2_ production in the thermal decomposition of TNTA at high temperatures are expected.

## 4. Conclusions

We report predictions using quantum mechanics of the most stable crystal structures of TNTA, which is a promising green high-energy-density material. Firstly, the partial RESP charges were used with the COMPASS force field to find the main nonbonded interactions which may affect the space groups. Then, unit cell parameters, Hirshfeld surface analysis, and fingerprint plots of the seven space groups were used to predict the most promising space groups and packings. Finally, the thermal decomposition of TNTA was researched by reactive molecular dynamics using ReaxFF-lg. We found that the Pna2_1_ space group with the corresponding cell parameters of Z = 4, a = 8.69 Å, b = 9.07 Å, c = 9.65 Å, and α = β = γ = 90.0° was the most compatible with TNTA. The pyrolysis mechanism implies that increasing the temperature can accelerate the decomposition rate of TNTA. The two main products, CO_2_ and N_2_, had the same variation trend, and the amount of N_2_ was slightly higher than that of CO_2_ during the whole reaction time.

## Figures and Tables

**Figure 1 materials-15-03873-f001:**
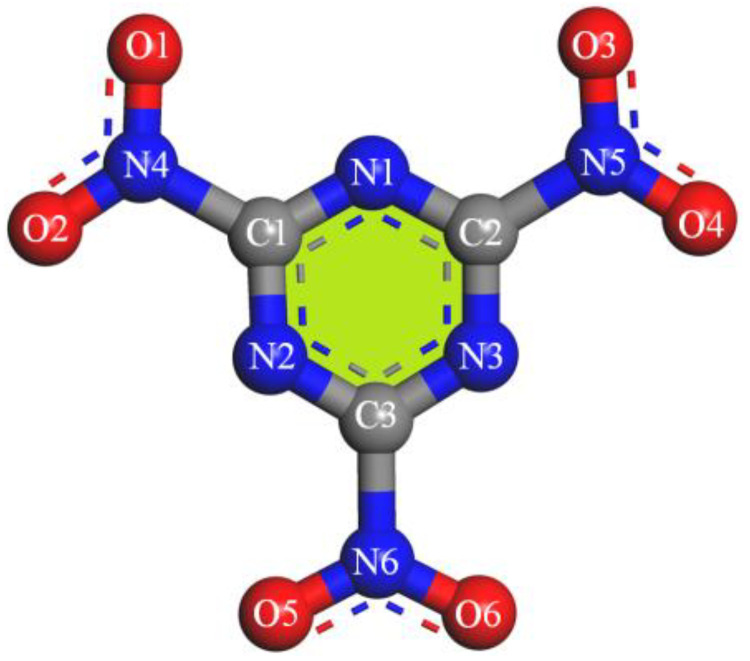
Structure and atomic numbering of TNTA.

**Figure 2 materials-15-03873-f002:**
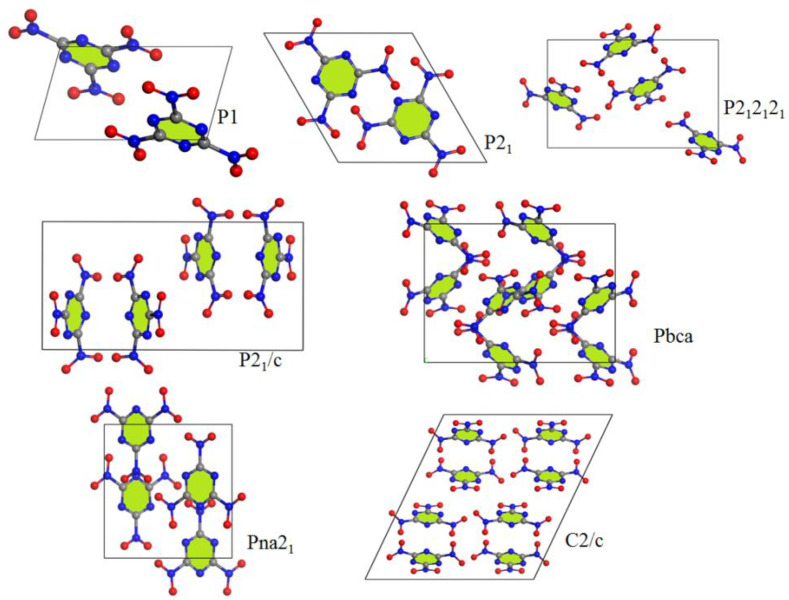
Most stable structures of TNTA for 7 different space groups obtained using Monte Carlo simulated annealing and Dreiding FF optimization.

**Figure 3 materials-15-03873-f003:**
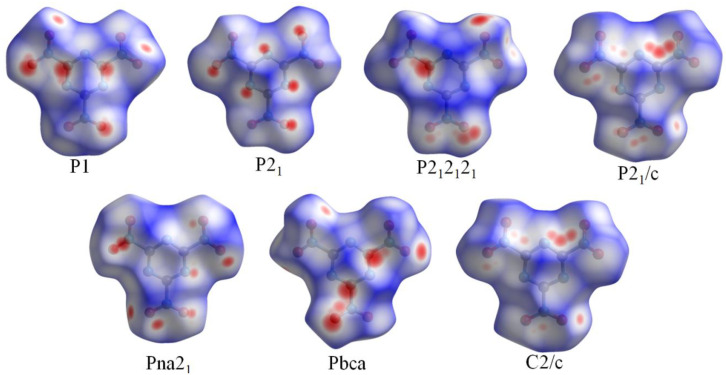
Hirshfeld surfaces of seven space groups in crystal stacking.

**Figure 4 materials-15-03873-f004:**
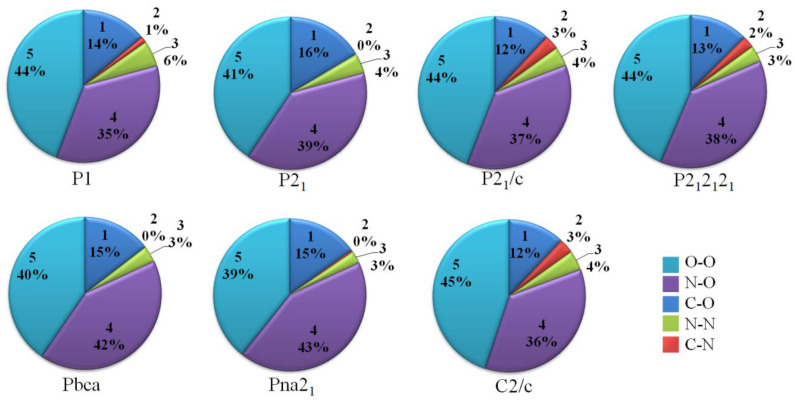
The individual atomic contact percentage contributions to the Hirshfeld surface of seven space groups.

**Figure 5 materials-15-03873-f005:**
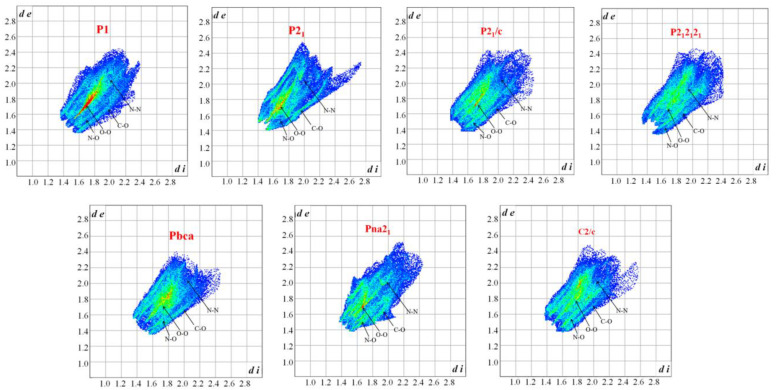
Two-dimensional fingerprint plots in crystal stacking.

**Figure 6 materials-15-03873-f006:**
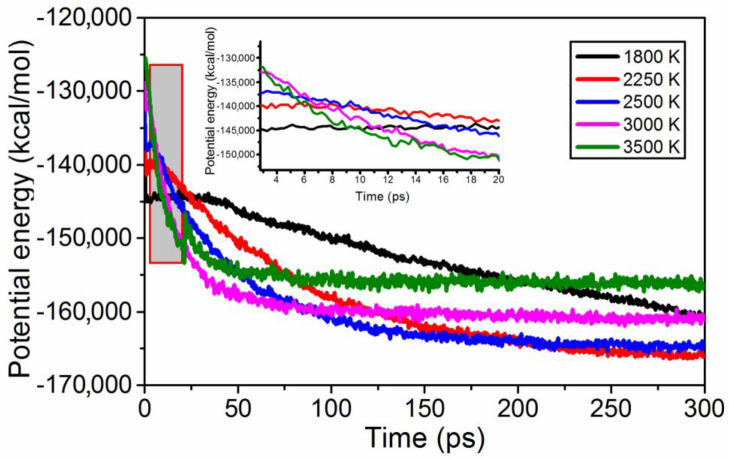
Evolution of the potential energy of the system over time at different temperatures.

**Figure 7 materials-15-03873-f007:**
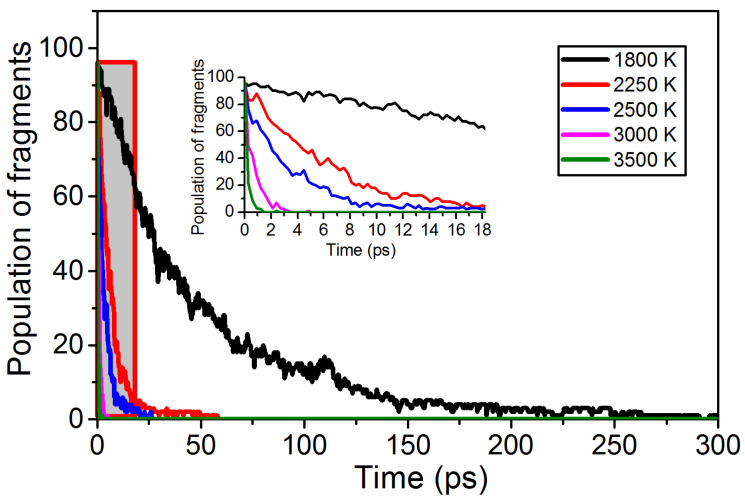
The population of TNTA molecules over time during the whole decomposition stage at the five extreme conditions.

**Figure 8 materials-15-03873-f008:**
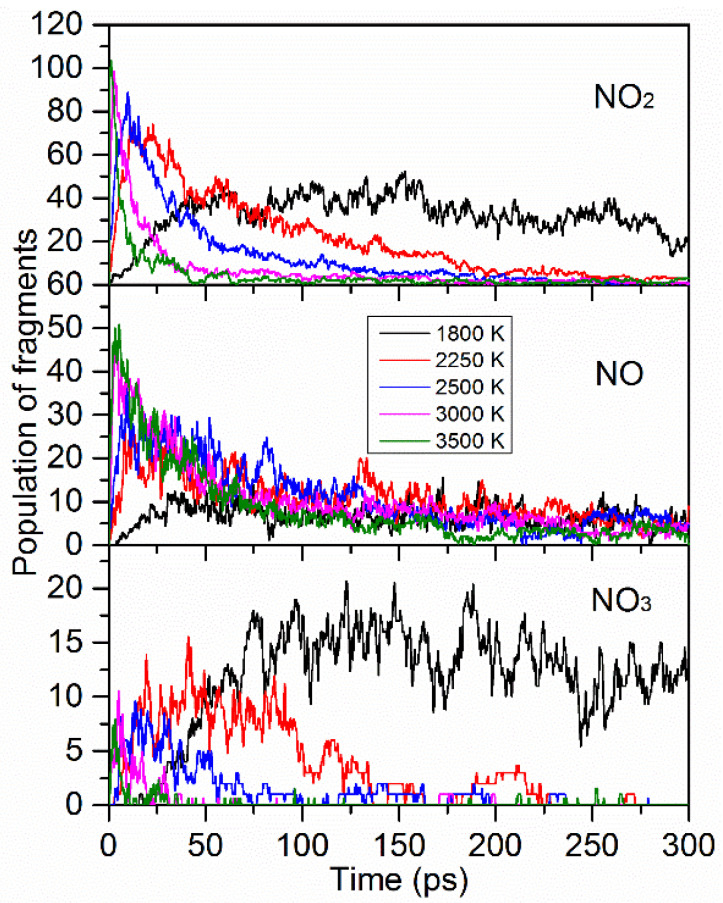
Time evolution of the main fragments NO_2_, NO, and NO_3_ during the whole decomposition stage at the five extreme conditions. The thick trendlines correspond to the actual concentration data of the matching color.

**Figure 9 materials-15-03873-f009:**
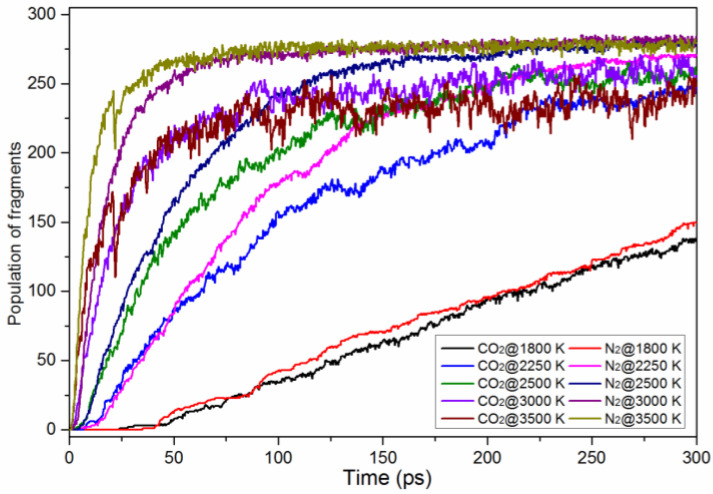
Time evolution of the main products, CO_2_ and N_2_, during the whole decomposition stage at the five extreme conditions. The thick trendlines correspond to the actual concentration data of the matching color.

**Table 1 materials-15-03873-t001:** Atom names, force field parameters, and RESP charges taken from COMPASS for prediction of crystal morphology.

	Force Field Types	Atom Type Description	RESP Charges
C1–C3	C_3_N	sp^2^, double bond to N (-C=N-)	0.78
O1–O6	O_12_	sp^2^, in nitro group (-NO_2_)	−0.35
N1–N3	N_2=_	sp^2^, double bond to C (-N=C-)	−0.73
N4–N6	N_3_O	sp^2^, in nitro group (-NO_2_)	0.65

**Table 2 materials-15-03873-t002:** Unit cell parameters of the possible molecular packing of TNTA.

Method/ Functional	Space Groups	Z	E (kJ/mol)	ρ (g/cm^3^)	a (Å)	b (Å)	c (Å)	α (°)	β (°)	γ (°)
GGA-PBE	P1	2	−3.82	1.85	10.99	5.51	9.02	86.84	116.27	124.64
	P2_1_	2	−3.72	1.83	9.06	5.48	9.12	90.0	60.03	90.0
	P2_1_2_1_2_1_	4	−3.87	1.83	14.12	6.26	8.90	90.0	90.0	90.0
	P2_1_/c	4	−3.82	1.86	8.15	15.65	6.42	90.0	109.16	90.0
	Pna2_1_	4	−4.11	1.89	8.69	9.07	9.65	90.0	90.0	90.0
	Pbca	8	−3.91	1.90	13.17	9.57	12.01	90.0	90.0	90.0
	C2/c	8	−3.88	1.84	15.70	6.48	17.04	90.0	64.49	90.0
GGA-PW91	P1	2	−4.02	1.76	10.07	6.07	8.89	90.14	117.46	118.00
	P2_1_	2	−3.96	1.73	9.16	5.68	9.21	90.0	59.79	90.0
	P2_1_2_1_2_1_	4	−4.05	1.75	14.29	6.33	9.09	90.0	90.0	90.0
	P2_1_/c	4	−4.01	1.75	8.30	15.91	6.59	90.0	109.57	90.0
	Pna2_1_	4	−4.24	1.80	8.73	9.16	9.98	90.0	90.0	90.0
	Pbca	8	−4.07	1.78	13.55	9.71	12.24	90.0	90.0	90.0
	C2/c	8	−0.87	1.76	15.80	6.58	17.40	90.0	64.70	90.0
	P1	2	22.63	2.28	9.21	5.39	8.40	90.65	120.99	113.80
LDA-CA-PZ	P2_1_	2	23.01	2.23	8.73	4.89	8.72	90.0	59.88	90.0
	P2_1_2_1_2_1_	4	22.32	2.30	13.36	5.61	8.33	90.0	90.0	90.0
	P2_1_/c	4	22.49	2.28	7.71	14.02	6.16	90.0	108.96	90.0
	Pna2_1_	4	22.17	2.31	8.30	8.58	8.70	90.0	90.0	90.0
	Pbca	8	22.32	2.32	12.13	8.96	11.37	90.0	90.0	90.0
	C2/c	8	−35.8	1.77	15.79	6.53	17.45	90.0	64.57	90.0

**Table 3 materials-15-03873-t003:** Evolution details of various species.

Species	Appearance Time (ps)			Life Time (fs)			Max Numbers		
	1800 K	2500 K	3500 K	1800 K	2500 K	3500 K	1800 K	2500 K	3500 K
NO_2_	0.3	0.3	0.3	2616	2384	1343	53	91	106
NO	4.2	0.3	0.3	1032	1160	1090	17	39	53
NO_3_	15.9	3.3	0.6	4629	1771	568	21	10	9
CO_2_	24	2.1	0.9	17,561	14,017	4494	138	265	256
N_2_	35.1	4.2	1.2	48,558	25,722	11,026	150	282	284

**Table 4 materials-15-03873-t004:** Elementary reactions and net flux (NF).

2500 K			3500 K		
No.	Reactions	NF	No.	Reactions	NF
R0	CO_2_N_2_ → N_2_ + CO_2_	1053	R0	CO_2_ + N_2_ → CO_2_N_2_	3116
R1	N_2_ + CO_2_ → CO_2_N_2_	1050	R1	CO_2_N_2_ → CO_2_ + N_2_	3103
R2	C_2_O_4_ → 2 CO_2_	531	R2	C_2_O_4_ → 2 CO_2_	2590
R3	CO_3_N → CO_2_ + ON	499	R3	2 CO_2_ → C_2_O_4_	2539
R4	2 CO_2_ → C_2_O_4_	499	R4	CO_2_ + CO → C_2_O_3_	604
R5	CO_2_ + ON → CO_3_N	498	R5	C_2_O_3_ → CO_2_ + CO	599
R6	CO_4_N → CO_2_ + O_2_N	260	R6	CO_2_ + ON → CO_3_N	454
R7	CO_2_ + O_2_N → CO_4_N	254	R7	CO_3_N → CO_2_ + ON	448
R8	ON_3_ → N_2_ + ON	251	R8	N_4_ → 2 N_2_	400
R9	N_2_ + ON → ON_3_	245	R9	2 N_2_ → N_4_	394
R10	2 ON → O_2_N_2_	197	R10	CON_2_ → N_2_ + CO	337
R11	O_2_N_2_ → 2 ON	175	R11	N_2_ + CO → CON_2_	327
R12	2 O_2_N → O_4_N_2_	149	R12	CO_2_ + C_2_O_3_ → C_3_O_5_	294
R13	CO_2_ + CO → C_2_O_3_	143	R13	ON_3_ → N_2_ + ON	292
R14	CON_2_ → N_2_ + CO	142	R14	N_2_ + ON → ON_3_	291
R15	O_4_N_2_ → 2 O_2_N	128	R15	C_3_O_5_ → CO_2_ + C_2_O_3_	279
R16	C_2_O_3_ → CO_2_ + CO	122	R16	C_2_O_4_ + CO_2_ → C_3_O_6_	197
R17	N_2_ + CO → CON_2_	121	R17	C_3_O_6_ → C_2_O_4_ + CO_2_	189
R18	CON + ON → CO_2_N_2_	114	R18	CO_2_ + CO_3_ → C_2_O_5_	178
R19	N_2_ + CON → CON_3_	111	R19	N_2_ + C_2_O_3_ → C_2_O_3_N_2_	166
R20	CON_3_ → N_2_ + CON	108	R20	C_2_O_3_N_2_ → N_2_ + C_2_O_3_	164
R21	C_2_O_3_N → CO_2_ + CON	98	R21	CO_2_ + O_2_N → CO_4_N	152
R22	O_3_N_2_ → O_2_N + ON	92	R22	C_2_O_5_ → CO_2_ + CO_3_	142
R23	CO_2_N_2_ → CON + ON	85	R23	CO_4_ → CO_2_ + O_2_	134
R24	O_2_N + ON → O_3_N_2_	84	R24	C_2_O_4_N_2_ → 2 CO_2_ + N_2_	134
R25	CO_2_ + CON → C_2_O_3_N	83	R25	CO_4_N → CO_2_ + O_2_N	131
R26	O_2_N_2_ + ON → O_3_N_3_	79	R26	C_2_O_4_ + CO_2_ → C_2_O_4_ + CO_2_	131
R27	C_3_O_6_N_6_ → O_2_N + C_3_O_4_N_5_	78	R27	C_3_O_6_ → 3 CO_2_	127
R28	CO_2_ + C_2_O_3_ → C_3_O_5_	78	R28	C_2_O_4_N_2_ → C_2_O_4_ + N_2_	124
R29	C_3_O_5_ → CO_2_ + C_2_O_3_	77	R29	CO_2_ + O_2_ → CO_4_	122
R30	N_4_ → 2 N_2_	76	R30	2 CO_2_ + N_2_ → C_2_O_4_N_2_	121

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
