# Peer review of "Theoretical Prediction of Structures and Properties of 2,4,6-Trinitro-1,3,5-Triazine (TNTA) Green Energetic Materials from DFT and ReaxFF Molecular Modeling"

_materials, 2022, doi:10.3390/ma15113873_

Round 1

Reviewer 1 Report

The manuscript by Zhou Mingming and Dong Xiang describes a computational study devoted to possible crystal structures of 2,4,6-trinitro-1,3,5-triazine (TNTA). The synthesis has not succeded yet, and the study predicts properties of crystal structures and its decomposition (which is important in the case of high-energy systems. The study is well-written, the methodology is correct and I think that the manuscript should be published in Materials. However, there are some minor issues that should be corrected:
1) line 9: This sentence suggests that nitryl cyanide has not been reported yet which is not true [M. Rahm, G. Belanger-Chabot, R. Haiges, K. O. Christe, Angew. Chem. Int. Ed. 2014, 53, 6893 –6897]. Moreover if the Authors refer to nitryl cyanide, its synthesis should be mentioned in the introduction section as the syntheisis is very important and for years posed a huge synthetic challange.
2) line 62: "Fig. 1" shoudl be changed to "Figure 1" as all other figures are mentioned in a "Figure X" manner.
3) Either at the end of 3.1 section or at the beginning of 3.2 section it should be clearly stated why MD simulations were performed solely for the crystal structure with the Pna21 space group. 

Reviewer 2 Report

In this manuscript different packing structures of TNTA are investigated by force field and DFT simulations. The authors determine which of previously proposed structures are in best accordance with RESP charges. On this basis, the  Pna21 crystal structure is predicted to be most likely. In the second part of the manuscript the decomposition pathways and its temperature dependence are studied through MD simulations.

From the first few references it seems that the structure and properties of TNTA have already been studied by theoretical means. Neither the abstract nor the introduction clearly distinguish previous work from the new studies in the manuscript. It should be made clear in particular what has been done (or not) experimentally and what theoretically.

This issue is further enhanced by the language. Some phrasings are rather difficult to understand and the English should be checked carefully before publishing to avoid misunderstandings and misleading the reader. E.g. the first sentence in the abstract reads “The nitryl cyanide, O2NCN, as a new high-energy molecule, has not yet been reported.”, while the first sentence in the introduction states that “ab initio theory was (corr. has been) used to assess the properties … of TNTA in 1907.” This is virtually impossible. Further, in computational details it is written that “The original unit cell obtained from the experiment was expanded 62 into a 3*4*2” raising questions about previous experimental work on this system.

Computational approach: for the optimization of dimers and trimers it is advisable to use DFT with dispersion correction (e.g. B3LYP-D3) along with diffuse functions in the basis set (something like 6-311g(d,p)++). It is unclear if the authors did a molecular structure optimization or an optimization of the crystal with periodic boundary conditions (in combination with simulated annealing) . Later in the discussion (line 114) the authors state “Here we applied the PBE and PW91 pseudopotential to GGA and the PBE pseudopotential to LDA functional to the input structures” in connection with Table 2. PBE is not a LDA functional and it should be made clear where exactly the authors apply those pseudopotentials in their force field study.

The molecule seems to change name to TATN somewhere in the discussion (line 167). Check TATN vs. TNTA in the paper. Introduce abbreviations (HMX, RESP, …)

The MD study is interesting and seems comprehensible. Since this is not within my area of expertise, I cannot comment on the novelty of these findings or the methodology.  

Overall, I would recommend publication of this manuscript in this journal after the authors address the points above.
